# Do Preoperative Corticosteroid Injections Increase the Risk of Infection after Shoulder Arthroscopy or Shoulder Arthroplasty? A Systematic Review

**DOI:** 10.3390/healthcare12050543

**Published:** 2024-02-24

**Authors:** Ludovico Lucenti, Flora Maria Chiara Panvini, Claudia de Cristo, Damiano Rapisarda, Marco Sapienza, Gianluca Testa, Vito Pavone

**Affiliations:** Department of General Surgery and Medical Surgical Specialties, Section of Orthopaedics and Traumatology, Policlinico Rodolico-San Marco, University of Catania, 95123 Catania, Italy; ludovico.lucenti@gmail.com (L.L.);

**Keywords:** PJI, preoperative corticosteroid injection, periprosthetic infection, arthroscopy, arthroplasty, shoulder

## Abstract

Introduction: Corticosteroid injections have demonstrated short-term benefits for shoulder pain. This symptomatic treatment method is used in various inflammatory conditions that affect the shoulder joint. Corticosteroid joint injections are not without risks and complications. Adverse effects have been documented, including damage to the articular cartilage, tendon rupture, and attenuation of the subject’s immune response. The aim of this study was to examine the timing of preoperative corticosteroid injections on infectious outcomes of shoulder arthroscopies and shoulder arthroplasty. Materials and Methods: In accordance with the guidelines of the Preferred Reporting Items for Systematic Reviews and Meta-Analyses (PRISMA), the PubMed, Cochrane, and Science Direct databases were systematically reviewed by two independent authors in January 2024. After full-text reading and checking the reference lists, 11 article were included. Results: Patients who received a shoulder injection within three months prior to undergoing shoulder arthroplasty exhibited a markedly elevated incidence of infection. In addition, a significantly increased risk of periprosthetic joint infection (PJI) at 90 days postoperatively in patients who received CSIs (corticosteroid injections) within 1 month prior to shoulder arthroplasty was found. Different authors consider CSI injections within the 2 weeks prior to shoulder arthroscopy surgery principally associated with an increased risk of postoperative infection. Discussion: There is still no consensus on the correct timing of preoperative CSIs in both arthroscopic and arthroplasty procedures. The literature does not identify whether the number of preoperative injections could increase the risk of periprosthetic infection. Obesity, sex, and smoking did not have a significant effect on PJIs; alcohol abuse could be considered as a risk factor for PJIs with CSIs. Both in prosthetic surgeries and in arthroscopy procedures, modifiable and unmodifiable factors play secondary roles. The risk of postoperative infection is greater within 3 months, although it is almost comparable at one- and two-year follow-ups.

## 1. Introduction

Corticosteroid injections (CSIs) have demonstrated short-term benefits for various inflammatory conditions that affect the shoulder joint [1]. These conditions include osteoarthritis, adhesive capsulitis, rheumatoid arthritis, rotator cuff tendinosis, subdeltoid bursitis, and impingement syndrome [2]. Corticosteroids effectively disrupt the inflammatory and immune cascade at various stages, offering pain relief. However, despite the documented short-term benefits, corticosteroid joint injections carry inherent risks and may lead to complications [3]. In some anatomical areas, preoperative corticosteroid injections have occasionally shown adverse postoperative effects [4]. For instance, injections administered before total knee and hip arthroplasty have been linked to an increased risk of postoperative infection [5,6]; other effects of corticosteroid injections are impaired wound healing, surgical site infection, tendon rupture, damage to the articular cartilage, attenuation of the subject’s immune response, skin hypopigmentation, fatty atrophy, and hyperglycemia [7]. Corticosteroid injections administered within three months prior to surgery have been demonstrated to elevate infection rates [8]. 

Periprosthetic joint infection (PJI) in the shoulder joint is an infrequent yet severe complication of shoulder arthroplasties. The average occurrence has been documented at 1.1%; following reverse arthroplasty, it may rise to 3.8% and escalate to 10% in the subset of young male patients who undergo a reverse prosthesis [9]. 

Periprosthetic infections are commonly associated with microorganisms, notably skin pathogens such as Staphylococcus species and Cutibacterium acnes (formerly known as Propionibacterium acnes). Recent research underscores that Cutibacterium acnes is identified in a significant percentage, ranging from 31% to 70% in periprosthetic shoulder infections. This higher incidence in shoulder arthroplasties compared to other joints is likely influenced by the proximity of the surgical site to the axillary region [10]. Factors contributing to the risk of periprosthetic shoulder infections include post-traumatic osteoarthritis, prior surgeries, recurrent corticosteroid injections, systemic corticosteroid therapy, other immunosuppressive medications, rheumatoid arthritis, and diabetes mellitus [11]. 

Shoulder arthroscopy is a common orthopedic procedure employed for various shoulder conditions. The reported incidence of surgical site infection after shoulder arthroscopy ranges from zero to 3.4%. The relatively low incidence of infections makes it difficult to identify if it has posed a challenge in numerous studies, as they often lack the statistical power needed to identify independent patient-related risk factors linked to postoperative infections following shoulder arthroscopy [12]. Significant contributors to the risk of infection after shoulder arthroscopy include factors such as revision surgery, intraoperative and preoperative steroid injection, anemia, malnutrition, age, male gender, obesity, depression, and others [13].

The purpose of this study was to examine the timing of preoperative corticosteroid injections on infectious outcomes of shoulder arthroscopies and shoulder arthroplasties.

## 2. Materials and Methods

### 2.1. Study Selection

In accordance with the guidelines of the Preferred Reporting Items for Systematic Reviews and Meta-Analyses (PRISMA) [14], the PubMed, Cochrane, and Science Direct databases were systematically reviewed by two independent authors (FP and DR). Two different research strings were used, as follows: “((preoperative injection) AND (arthroscopy) AND (shoulder))” and “((preoperative injection) AND (arthroplasty) AND (shoulder)). From each included original article, a standard data entry form was utilized to extract the number of patients, type of study, treatment, follow-up, and year of the study. 

The study quality assessment was performed in duplicate by two independent reviewers (FP and DR). Discussing conflicts about data were resolved via a consultation with a senior surgeon (LL). 

### 2.2. Inclusion and Exclusion Criteria

Eligible studies for this systematic review included arthroplasty or arthroscopy procedures in patients treated with preoperative injections (corticosteroid injections (CSIs) or hyaluronic acid (HA)) involving only the shoulder joint. Articles were selected in the English language, evaluating the recent 10 years of research (2013–2023). All articles that focused on the main topic found in the literature in other languages, or were published before 2013, were excluded.

### 2.3. Risk of Bias Assessment

The study data extracted were author, country, year of publication, study type, sample size, demographics, number of injections, timing of injection, surgical procedure. Bias was assessed using the Risk of Bias in Non-Randomized Studies of Interventions (ROBINS-I) tool [15]. Two authors (FP and DR) performed the evaluation independently. Any discrepancy was discussed with the senior investigator (LL) for the final decision.

## 3. Results

### 3.1. Included Studies

A total of 384 articles were found. We included the recent 10 years of research on the main topic (2013–2023) in our study; for this reason, 34 articles were excluded. After the exclusion of duplicates, 342 articles were selected. At the end of the first screening, following the previously described selection criteria, we selected 40 articles eligible for full-text reading. Meta-analyses or systematic reviews were excluded from this study. We preferred to evaluate just the articles that had a follow-up period of at least of 6 months. Ultimately, after full-text reading and checking the reference lists, we selected 11 articles, following previously outlined criteria. A PRISMA [16] flowchart of the method of selection and screening is provided in Figure 1. The main findings of the included articles are summarized in Table 1. 

### 3.2. Articles

All articles [16,18,19,20,21,22,23,24,26] included in this study focus on preoperative injections and their plausible relation with postoperative infection involving the shoulder joint.

From our analysis, the systematically selected articles were analyzed and divided into two groups. For greater ease, we distinguish the “arthroplasty” group and the “arthroscopy” group. None of the studies in the literature eligible for this systematic review considered HA injections.

#### 3.2.1. Shoulder Arthroplasty

Stadecker et al. [24], in their retrospective cohort study on large databases, divided patients according to the timing of preoperative steroid injections, with intervals of 1–3 months (m), 3–6 m, 6–9 m, and 9–12 m. They evaluated the presumed association, if presented, with modifiable and non-modifiable risk factors: age, sex, smoke, and BMI. Evaluating long-term outcomes (2 years), patients who received an injection within three months before total shoulder arthroplasty (TSA) or reverse shoulder arthroplasty (rTSA), demonstrated an elevated likelihood of undergoing all-cause revision surgery post-arthroplasty. Among the revisions, more than 55% were attributed to PJIs, while around 43% were due to non-infective causes. Within the cohort receiving injections less than three months prior, more than 70% of revisions were PJI-related, contrasting with the almost 30% that were attributed to non-infective causes. In the three- to six-month injection cohort, 60.0% of revisions were linked to PJIs, as opposed to 40.0% due to non-infective causes. The six- to nine-month injection cohort demonstrated that half of the revisions were associated with PJIs, and in the nine- to twelve-month injection cohort, only one third were secondary to PJIs.

Obesity, sex, and smoking did not exert a significant effect on PJI rates, aligning with the findings of Werner et al. [26] and Baksh et al. [19]. Notably, alcohol abuse emerged as a risk factor for PJIs with corticosteroid injections. Chronic kidney disease (CKD) and depression were also identified as significant risk factors, while diabetes mellitus, hypothyroidism, and tobacco use did not exhibit a significant association with the risk of postoperative infections.

In their study, Werner et al. [26] evaluated the risk of postoperative infections in patients who underwent corticosteroid injection at 3 months before surgical treatment and less than 3 months before surgical treatment. Patients who underwent shoulder arthroplasty within three months of receiving a shoulder injection exhibited a significantly higher incidence of infection after a 3-month and 6-month period when compared to the matched controls. However, no significant differences were observed in the incidence of postoperative infections between patients who underwent shoulder arthroplasty between 3 and 12 months after injection and their matched controls.

Baksh et al. [19] analyzed the incidence of postoperative infections in reverse shoulder arthroplasty. They found an increased risk of postoperative infections in patients who underwent preoperative CSIs < 1 month prior to rTSA (follow-up at 90 days, 1 y, and 2 y).

Patients who underwent reverse shoulder arthroplasty (rTSA) and received CSIs within one month before the procedure faced a significantly increased risk of PJIs at 90 days postoperatively. Furthermore, this heightened risk persisted at the 1-year mark after rTSA. Baksh et al. also analyzed the incidence of PJIs in TSA [18]. A significantly increased risk was identified at both 1-year and 2 years postoperatively for patients who received CSIs within four weeks of TSA.

In both studies [18,19], contrary to Werner et al. [26] and to Stadecker [24], they did not find any difference regarding the infection rate in patients who received CSIs > 1 month prior to rTSA.

On the other hand, in just one study, Rashid et al. did not find any statistical difference in their retrospective analysis on the rates of superficial surgical site infections (SSSIs) and deep surgical site infections (DSSIs) between patients who received a CSI approximately 11.4 months (range 2.5 months to 172.5 months) before surgery and the ones that did not. They believed that there was no statistically significant relation (*p* > 0.05) between preoperative CSIs and the development of post-arthroplasty infectious complications.

#### 3.2.2. Shoulder Arthroscopy

Werner et al. [26] observed a significantly higher incidence of infection in patients undergoing shoulder arthroscopy within three months of shoulder injection, both at 3 months and 6 months, when compared to the matched controls. Notably, no significant differences were found in the incidence of postoperative infection among patients undergoing shoulder arthroscopy between 3 and 12 months after injection and their matched controls.

In the study by Livesey et al. [16], CSIs within two weeks before surgery were correlated to a high risk of infection at 90 days post-surgery. Also, CSIs administered 2–4 weeks before surgery were linked to a high risk of infection at 90 days, 1 year, and 2 years.

They identified alcohol, diabetes, and tobacco as additional risks factors at 90 d, 1 y, and 2 y after shoulder arthroscopy.

Remily et al. [23] showed similar results in a patient underwent CSIs 1–2 w and 2–4 w prior to arthroscopy. In their study, CSIs > 4 weeks before surgery were not associated with an increased risk of infection.

In the study by Forsythe et al. [21], it was observed that patients who received a shoulder injection within one month before their surgical procedure exhibited a high risk of infection. However, there was no significant increase in the risk of infection when patients received a shoulder injection more than one month before surgery. Notably, independent risk factors for infection included male sex, obesity, diabetes, smoking, and the administration of preoperative corticosteroid injections within one month before the surgical procedure.

Bhattacharjee et al. [20] showed again a significant difference if the injection was received within 2 weeks before surgery (6.33% vs. 0.55%, *p* < 0.0001).

Significantly increased odds of revision in rotator cuff repair (RCR) were found in patients who received at least one injection of steroids in the year prior to surgical treatment.

In their research, Weber et al. [25] stated that the cumulative revision rate was almost 5% for individuals who underwent injections, contrasting with almost 3% for those who did not have preoperative injections.

Even though open surgeries can have a high risk of infection rates, a total of 22,375 shoulders that underwent steroid injections were analyzed. The aggregate revision rate, without considering the type of surgery, exhibited a notable disparity, standing at almost 5% for patients subjected to injections in contrast to a 3.0% rate among those who abstained from such treatment. The authors reported that the maximal tendency for necessitating a revision, manifested when there was temporal separation between the administered injection and subsequent RCR, was limited to a single month. The probability of revision exhibited a discernible decrement as the temporal interval between injection and RCR extended. The authors noticed that a singular injection engendered an increased odds ratio for revision subsequent to the index RCR (OR, 1.25; 95% CI, 1.10–1.43; *p* < 0.0001), while patients exposed to two or more injections confronted a more than twofold augmentation in the odds of undergoing a revision RCR following the initial procedure (OR, 2.12; 95% CI, 1.82–2.47; *p* < 0.0001). There existed no statistically significant difference in the reoperation rate between patients who received an injection within one year preoperatively and the control cohort.

Agarwalla et al. [17] assert that patients who received preoperative injections demonstrated markedly elevated rates of reoperation, encompassing revision rotator cuff repair and subacromial decompression, within the 6- to 12-month post-index surgical period. A salient observation from their study was that the reoperation rate within the 3 to 6 months following rotator cuff repair was notably higher among patients previously subjected to a preoperative injection, with no observable differences in the initial 3 months postoperatively.

## 4. Discussion

In clinical practice, intra-articular injections are frequently employed to manage symptoms in patients afflicted by shoulder osteoarthritis. The utilization of CSIs and the temporal relationship of their administration in relation to shoulder arthroplasty hold potential implications for surgical outcomes, notably an elevated risk of PJIs. Some investigations have documented an infection rate of 0.27% following shoulder arthroscopy and a significant 15% infection rate following shoulder arthroplasty [27,28]. The escalating incidence of post-arthroplasty infections is emerging as a substantial concern, particularly considering the rising prevalence of antibiotic-resistant organisms and instances of failed arthroplasty with positive cultures [29].

From the analysis conducted in this systematic review, there is still no consensus on the correct timing of preoperative CSIs in both arthroscopy and arthroplasty procedures. Different authors suggest that the greatest risk of PJIs due to preoperative injections occurs in the first 3–6 months postop if the injection is performed within 3 months before surgical treatment [24,26]. The incidence of PJIs compared to aseptic causes is similar at a 1- and 2-year follow-up in patients who received a preoperative injection and those who did not.

Baksh et al. [19] even argue that the risk increases if the injection is performed in an even closer window (<1 month). No difference is found if the injection is performed >1-month preop. Baksh et al. [18,19] believe that TSA and rTSA have a comparable risk of infection if the preoperative injection is performed within 1 month prior to the surgical procedure. No differences are found if the injection is performed two months or more before prosthetic treatment. This major difference could be due to the different infiltrative treatment windows (Table 2A,B).

Rashid et al. [22] found no differences between the patients who received at least one injection and the ones who did not within a year before surgery, but this could be due to the fact that a low number of patients were analyzed. This study was significantly constrained by limitations in statistical power and the considerable variability in the extended intervals between injection(s) and subsequent arthroplasty. These factors substantially impeded this study’s capacity to discern and establish any potential associations.

The literature does not identify whether the number of preoperative injections could increase the risk of periprosthetic infections. Many articles selected for this systematic review analyzed national databases. In fact, despite the huge sample analyzed, Baksh et al. [19] believed that an important limitation regarding this topic is the fact that national databases do not consider the differences in CSI techniques or surgeons’ proficiencies. They also do not know if sterile techniques were used for all CSIs. They also assumed that the injections were intra-articular in all procedures. Moreover, Stadecker et al. [24] could not differentiate glenohumeral versus subacromial injections due to the limits of existing coding terminology applied in national databases. Similarly to Baksh et al. [19], Werner et al. [26] considered the fact that they had no system to confirm the accuracy of the injections, which could differ in relation to the practice of the physician administering the injection and other aspects, such as the use of ultrasound guidance. Only Rashid et al. [22] defined the characteristics of the injection performed, which were carried out in a fluoroscopy room by radiologists after sterile preparation. In their study, an injection of 40 mg of methyl-prednisolone combined with 10 mL of 0.25% bupivacaine was added into the joint through a 22-gauge spinal needle.

Unlike the PJI group, Weber et al. [25] thought that a greater number of CSIs in the affected shoulder within 1-year preop could be related to an implemented risk of infections after an arthroscopic procedure.

There is greater consensus in the literature that the 0–30-day window before arthroscopic treatment may be the most favorable regarding the risk of post-procedural infections [21,23]. Livesey et al. [16] and Bhattacharjee et al. [20] agree that the risk of infections related to CSIs is major if performed 2 weeks prior to surgical treatment. The risk of postoperative infections is greater within 3 months, although it is almost comparable at the one-year follow-up. Both in prosthetic surgeries and in arthroscopy procedures, modifiable and unmodifiable factors play secondary roles [16,24,25]. Baksh et al. [19] found a greater correlation between alcohol, CSIs, and the increase in the risk of PJIs.

All evaluated studies did not mention the typology of the steroid injections, so we are not able to understand if those could be considered as a factor related to an increased risk of postoperative infections. As well as arthroplasty studies, some limitations in the arthroscopy group were found. Using national databases could provide a large cohort for study but could include a combination of shoulder arthroscopies with variable indications for surgical procedure and operative times [20]. Forsythe and colleagues [21] did not specify the exact location of the injection. It was not possible for them to distinguish patients who received an injection into the sub-acromial space or glenohumeral joint. Agarwalla and colleagues [17], agreeing with the other studies found in the literature, also considered that using a large database could be a strength but also a limitation. They tried to have a large sample and aimed to make it as homogeneous as possible.

The study group was compared to a control group matched for age, sex, diabetes, and smoking. Other important factors that may have an important impact, such as hand dominance and time between injury and surgery, were not considered.

The location and route of administration of the corticosteroids, relative to the surgical site, could be an important factor for the risk of infection [30], but low-quality studies were found in the literature regarding this.

According to Papvasillou et al. [7], it could be useful to distinguish superficial surgical site infections (SSSIs) and deep surgical site infections (DSSIs).

SSSI is an infection that can happen within 30 days after surgery and involves the area around the surgical incision, which occurs as either a positive wound swab and/or purulent discharge. This type of infection is typically managed through local debridement and antibiotic therapy. In contrast, DSSI is an infection that occurs within 6 months after surgery, characterized by purulent discharge, a positive swab, joint aspirate, or tissue biopsy, or the presence of pus cells during microscopy analysis. Additionally, it may manifest as an abscess or exhibit other signs of infection involving deep tissue. Staged reimplantation is often required for its management [31,32].

From the article analyzed, it was not possible to distinguish between SSSIs and DSSIs that occurred in patients who received CSIs and who did not and experience any infections after shoulder arthroplasty or arthroscopy procedures. Rashid et al. [22] consider this to be an important parameter but, in their sample, they did not find any differences between the CSI group and the control group; also, they did not find any infections after arthroplasty procedures in the two groups.

Studies conducted by Amin et al. [33] found no differences between the effects of preoperative HA and CSIs.

In their study, Richardson et al. [34] analyzed the occurrence of PJIs subsequent to both corticosteroid and hyaluronic acid injections in TKA. Their findings revealed an escalated PJI risk in both treatment groups, implying that the heightened risk may be attributed to the introduction of a foreign body rather than the anti-inflammatory and immunosuppressive effects specifically associated with corticosteroids. A similar conclusion was reached by Livesey and colleagues [16], who demonstrated that preoperative needle penetration, inclusive of procedures such as magnetic resonance arthrograms, was linked to an elevated risk of a postoperative infection when conducted within a span of two weeks preceding the surgical intervention.

From our analysis, we could not evaluate whether HA injections could increase the risk of postoperative infections regarding shoulder arthroplasty or arthroscopy procedures. In our analysis, the lack of information relative to HA injections in the shoulder joint could not permit us to perform a complete evaluation of this topic.

This study has some limitations.

We were not able to evaluate whether comorbidities could play an actual role in the risk of periprosthetic infections, due to the fact that the data analyzed were not uniform in all the mentioned articles. Comorbidities were not studied in the arthroscopic procedure group. Some authors have tried to underline various non-associations; however, the sample size of the cohort under study should be increased, and the evaluation of any associations should be the primary aim. As already mentioned, it was not possible to evaluate the HA injections due to the lack of information found in the literature.

We were not able to optimally analyze whether the risk of infection was related to the type and location of the injection; this was because the studies eligible for systematic review mostly considered large databases and were not able to retrospectively evaluate these data. More studies focusing on this topic should be completed. From the extracted data, we were not able to perform a meta-analysis.

## 5. Conclusions

Despite positive data regarding preoperative CSIs within one month before arthroplasty procedures, more studies need to be performed. Many studies agree on the greater safety of preoperative CSIs at least 3 months before surgical treatment. Based on the findings with the literature, arthroscopic procedures could be performed safely even after a month post-injection. More studies need to be carried out, especially with regard to the shoulder joint and the possible correlation between the number of injections, types of injections, and the risks of postoperative infections.

## Figures and Tables

**Figure 1 healthcare-12-00543-f001:**
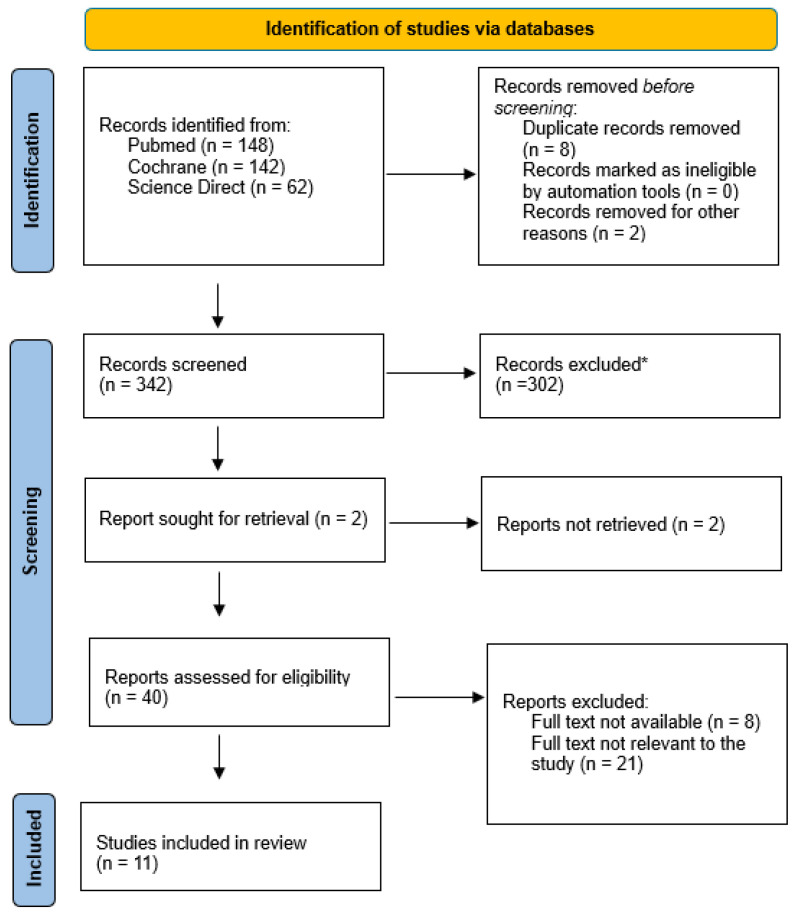
PRISMA 2020 flow diagram for new systematic reviews. * Abstract not suitable for research.

**Table 1 healthcare-12-00543-t001:** Summary of the eleven articles eligible for the systematic review. CSIs: corticosteroid injections. PJIs: periprosthetic joint infections. TSA: total shoulder arthroplasty. rTSA: reverse total shoulder arthroplasty. RCR: rotator cuff repair. d: day. m: month. y: year.

Author	Publ. Year	Number of Patients	Type of Study	Follow-Up	Results	Main Topic
Agarwalla et al. [17]	2019	12,054	Retrospective cohort, comparative study	3 m, 6 m, 9 m, 1 y	The rate of reoperation within 3 months of the index procedure was higher in the control group (3.7% vs. 3.1%, P 1⁄4 0.01); however, 3 to 6 months after the index procedure, the rate of reoperation was higher in patients who received an injection within 1 year of the index procedure (1.8% vs. 1.4%, P 1⁄4 0.03). The incidence of revision RCR (1.6% vs. 1.1%; odds ratio, 1.4; P 1⁄4 0.003) and incidence of subacromial decompression (1.5% vs. 1.1%; odds ratio, 1.3; P 1⁄4 0.01) 6 to 12 months after the index procedure were significantly higher in patients receiving an injection within 1 year before surgery.	Arthroscopy
Baksh et al. [18]	2023	1291	Retrospective cohort study	90 d, 1 y, 2 y	A significant increase in PJI risk at 1 year (odds ratio [OR] = 2.29, 95% confidence interval [CI] = 1.19–3.99, *p* = 0.007) and 2 years (OR = 2.03, CI = 1.09–3.46, *p* = 0.016) in patients who received CSIs within 1 month of TSA was noted. PJI risk was not significantly increased at any time point for patients who received a CSI greater than 4 weeks prior to TSA (all *p* ≥ 0.396).	Arthroplasty
Baksh et al. [19]	2023	20,898	Retrospective cohort study	3 m, 1 y	Significantly increased risk of PJIs at 90 days in patients who received CSIs within 1 month of rTSA (*p* < 0.001). Additionally, the PJI risk was increased at 1 year postoperatively in patients who received CSIs within 1 month of rTSA (*p* = 0.015). However, no significant increase in the PJI risk was noted at any time point for patients who received CSIs > 1 month before rTSA (all *p* ≥ 0.088).	Arthroplasty
Bhattacharjee et al. [20]	2019	4115	Retrospective cohort study	6 m	Significant increase in both the overall infection rate (*p* < 0.0001) and severe infection rate (*p* < 0.0001) in patients who received injections within 2 weeks before surgery (n = 79; 8.86% and 6.33%, respectively).	Arthroscopy
Forsythe et al. [21]	2019	12,060	Retrospective cohort study		There was no significant difference in the incidence of surgical site infection in patients receiving a shoulder injection at 0.7% compared with the control cohort at 0.8% (odds ratio [OR], 0.9 [95% confidence interval (CI), 0.7 to 1.1]; *p* = 0.2). However, patients receiving an injection within 1 month prior to operative management had a significantly higher rate of surgical site infection overall at 1.3% compared to the control group at 0.8% (OR, 1.7 [95% CI, 1.0 to 2.9]; *p* = 0.04).	Arthroscopy
Livesey et al. [16]	2023	3630	Retrospective cohort, prognostic study	3 m, 1 y, 2 y	Two main groups: CSI swithin 2 and 2/4 weeks of surgery. They were associated with an increased risk of postoperative infection. CSIs for 2 weeks at 90 days (OR, 1.72; *p* = 0.022), 1 year (OR, 1.65; *p* = 0.005), and 2 years (OR, 1.63; *p* = 0.002) following surgery. CSIs for 2–4 weeks increased the risk of postoperative infection at 90 days (OR, 1.83; *p* < 0.001), 1 year (OR, 1.62; *p* < 0.001), and 2 years (OR, 1.79; *p* < 0.001).	Arthroscopy
Rashid et al. [22]	2015	23	Retrospective cohort study	16.6 m	Patients received a CSI at approximately 11.4 months (range 2.5 months to 172.5 months) before their surgery. One patient developed a deep joint infection that warranted revision arthroplasty.	Arthroplasty
Remily et al. [23]	2023	9860	Retrospective cohort, prognostic study	3 m, 1 y, 2 y	Postoperative infection in patients who received CSIs 0–2 weeks before shoulder arthroscopy at 90 days (3.10, 95% confidence interval [CI] 1.62–5.57, *p* < 0.001), 1 year (2.51, 95% CI 1.46–4.12, *p* < 0.001), and 2 years (2.08, 95% CI 1.27–3.28, *p* = 0.002) compared to the control group. Patients who received CSIs 2–4 weeks before shoulder arthroscopy had a greater OR for infection at 90 days (2.26, 95% CI 1.28–3.83, *p* = 0.03), 1 year (1.82, 95% CI 1.13–2,82, *p* = 0.01), and 2 years (1.62, 95% CI 1.10–2.47, *p* = 0.012). Patients who received CSI safter 4 weeks had similar ORs of infection at 90 days (OR 1.15, 95% CI 0.78–1.69, *p* = 0.48), 1 year (OR 1.18, 95% CI 0.85–1.63 *p* = 0.33), and 2 years (OR 1.09, 95% CI 0.83–1.42, *p* = 0.54), compared to the control cohort.	Arthroscopy
Stadecker et al. [24]	2022	1632	Retrospective cohort study	2 y	On multivariate analysis, patients who received corticosteroid injection < three months prior to TSA or rTSA were at a significantly increased risk for revision (odds ratio (OR) 2.61 (95% confidence interval (CI) 1.77 to 3.28); *p* < 0.001) when compared to the control cohort. However, there was no significant increase in revision risk for all other timing interval cohorts.	Arthroplasty
Weber et al. [25]	2018	21,796	Therapeutic study	1 y	Patients who received injections prior to RCR were more likely to undergo RCR revision than the matched controls (odds ratio [OR], 1.52; 95% confidence interval [CI], 1.38–1.68; *p* < 0.0001). Patients who received injections closer to the time of index RCR were more likely to undergo revision (*p* < 0.0001). Patients who received a single injection prior to RCR had a higher likelihood of revision (OR, 1.25; 95% CI, 1.10–1.43; P 1⁄4 0.001). Patients who received 2 or more injections prior to RCR had greater than 2-fold odds of revision (combined OR, 2.12; 95% CI, 1.82–2.47; *p* < 0.0001) versus the control group.	Arthroscopy
Werner et al. [26]	2016	12,903	Multicenter study	3 m, 6 m	The incidence of infection after arthroscopy at 3 months (0.7%; odds ratio [OR], 2.2; *p* < 0.0001) and 6 months (1.1%; OR, 1.6; *p* = 0.003) was significantly higher in patients who underwent injection within 3 months before arthroscopy compared to the control group. The incidence of infection after arthroplasty at 3 months (3.0%; OR, 2.0; *p* = 0.007) and 6 months (4.6%; OR, 2.0; *p* = 0.001) was significantly higher in patients who underwent injection within 3 months before arthroplasty compared to the control group.	Arthroscopy and arthroplasty

**Table 2 healthcare-12-00543-t002:** (**A**) Group-specific selection considering period between injection and surgery: difference between the considered treatment windows in the arthroplasty group (y = year; m = month). (**B**) Group-specific selection considering period between injection and surgery: difference between the considered treatment windows in the arthroscopy group (w = week; m = month). The background color indicates the time period between injection and surgery.

(**A**)										
		**1 m**	**1–2 m**	**2–3 m**	**3–6 m**		**6–9 m**		**1 y**	**172.5 m**
Baksh et al. [18] (8)	2023									



Baksh et al. [19] (9)	2023									


Werner et al. [26] (17)	2016									

Rashid et al. [22] (13)	2015									
Stadecker et al. [24] (15)	2022									



(**B**)										
		**0–2 w**		**2–4 w**		**1–3 m**		**4–6 m**		**7–12 m**
Werner et al. [26] (17)	2016									

Weber et al. [25] (16)	2018									
Remily et al. [23] (14)	2023									

Agarwalla et al. [17] (7)	2019									
Bhattacharjee et al. [20] (10)	2019									



Forsythe et al. [21] (11)	2019									



Livesey et al. [16] (12)	2023									


## Data Availability

Not applicable.

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
