# Peer review of "Do Preoperative Corticosteroid Injections Increase the Risk of Infection after Shoulder Arthroscopy or Shoulder Arthroplasty? A Systematic Review"

_healthcare, 2024, doi:10.3390/healthcare12050543_

Round 1

Reviewer 1 Report

Comments and Suggestions for Authors

abstract: please specify the aim of study as it's not clear for readers

introduction: is too long , i suggest to shorten the introduction part

what method of quality assessement was used?  NHS? 

table legend 1 :  you wrote 9 article evaluated while it was 11 articles

- number of patients evaluated in each studies were high, however, i wondered why study of rachis et al was included with only 23 paitent,  how quality assesment was evaluated?

how authors confirm that intra-articular injection can increase PJI , and not due to co-morbidities of patiets (alcohol, diabetes, and tobacco, obesity , chronic renal failure etc ) ?   how these confounding bias were adressed , all these issue should be discussed 

was  injection location was searched in each study ? subacromial injeciton vs intra-articular injection , as risk of pji could be different  (as line 311-314) (forsythse et al ) 

Limitation of study should be mentioned in 1 paragraph including confounding bias, quality of studies  sepcially location of injection / co-morbidites etc

Reviewer 2 Report

Comments and Suggestions for Authors

Dear authors,

In this systematic review, you sought to examine the timing of corticosteroid injections on infectiooutcomes of shoulder arthroscopy and arthroplasty. 

Please find my comments below.

Abstract

Please note the following sentence needs to be fully revised.

Matherial and Methods: According to the guidelines of  the Preferred Reporting Items for Systematic Reviews and Meta-Analyses (PRISMA), multiple databases were systematically reviewed.

First of all, you have to determine which databases were searched and when. Also, typo errors must be corrected.

Introduction: Please refrain from single-sentence paragraphs. For a more appropriate presentation, I would advise you consider dividing this section into 3-4 separate paragraphs only.

Figure 1. Please note this figure is incomplete and incorrect. Please follow PRISMA flow chart and present a more comprehensive study selection process. Also, I am concerned about the limited number of articles you identified. In general, approximately 1000 record need to be screened for a high-quality systematic review.

I am also concerned about the fact that you only included papers published in the last ten years. As far as I know steroid injection has been widely used for decades and I see no reason why you have to rule out older studies (if any).

Comments on the Quality of English Language

English improvement is needed.

Author Response

Please see the attchment

Reviewer 3 Report

Comments and Suggestions for Authors

This is a well-executed systematic review. There are some minor comments that need to be addressed before recommending publication. 

1. It was obviously not possible to conduct a meta-analysis. Or was this not attempted? Please provide an explanation in the limitations. 

2. Then at least do a risk of bias and a publication bias assessment. That should be possible. 

3. There are minor ambiguities: e.g. the abbreviations of PSI and CSI are not explained in the abstract. 

4. Later in the introduction, the abbreviation CSI is not explained.

5. In the conclusion it says "CSI injection", which means: "corticosteroid injection injection".

6 Furthermore, it is not entirely clear to me what "HA" is. It is not explained. If it is hemiarthroplasty, what does "preoperative injection (CSI or HA)" mean in the methods under inlcusion exclusion criteria?

7. Check the manuscript for similar ambiguities.
